# Physical activity and sedentary behaviour counselling: Attitudes and practices of mental health professionals

Nipun Shrestha[1], Zeljko Pedisic[1], Danijel Jurakic[2], Stuart J. H. Biddle[3], Alexandra Parker[1,4]*

**1** Institute for Health and Sport, Victoria University, Footscray, Australia, **2** Faculty of Kinesiology, University of Zagreb, Zagreb, Croatia, **3** Physically Active Lifestyles Research Group (USQ-PALs), Centre for Health Research, University of Southern Queensland, Toowoomba, Australia, **4** Orygen, Centre for Youth Mental Health, University of Melbourne, Melbourne, Australia

* alexandra.parker@vu.edu.au

**Data Availability Statement:** The collected data are safely stored in the Victoria University data repository. Data contain potentially sensitive information about participants, and according to

## Abstract

### Background

Despite recent interest in the mental health benefits of increasing physical activity (PA) and reducing sedentary behaviour (SB), little is known about PA and SB counselling provided by mental health professionals. Therefore, the aim of this study was to explore the attitudes and practices of mental health professionals in recommending more PA and less SB to their clients.

### Methods

Quantitative data were collected using a modified version of the Exercise in Mental Illness Questionnaire in a sample of 17 Australian mental health professionals. The collected data were reported using percentages (for categorical data) and means and standard deviations (for numerical data). Additionally, in focus group discussions, 10 mental health professionals provided in-depth information about their clinical practice, facilitators, and perceived barriers in recommending more PA and less SB. They also provided suggestions on how to potentially improve their PA and SB counselling practices. The focus groups were audio-recorded, transcribed and analysed using thematic analysis.

### Results

Only 35.3% of participants have undergone formal training in recommending PA in the treatment of mental illness. Most participants (64.7%) ranked PA counselling among the top three types of mental health treatment. All participants reported recommending PA to their clients at least "occasionally", while 88% of them also provided SB counselling. However, the recommendations provided were usually not specific. The most commonly reported barriers for providing PA and SB counselling were a lack of knowledge and confidence. Participants also believed that, if they were more active themselves, they would be in a better position to recommend PA to their clients, by sharing their own experience of evidence-informed strategies designed to increase PA and reduce SB.

the ethics approval granted by the Victoria University Human Research Ethics Committee (VUHREC) data cannot be made publicly available. Controlled data access can be obtained upon a reasonable request submitted to the corresponding author or, in case of her absence, to the VUHREC (researchethics@vu.edu.au).

**Funding:** The authors received no specific funding for this work.

**Competing interests:** The authors have declared that no competing interests exist.

## Conclusion

The findings of this study indicate that mental health professionals commonly provide generic PA and SB counselling to their clients. PA and SB counselling in the mental health setting could be improved by: including training on PA and SB counselling in formal education and continued professional training for mental health professionals; implementing interventions to increase PA and reduce SB among mental health professionals themselves; and ensuring support from an exercise or PA promotion specialist as a part of a multi-disciplinary approach to mental health care.

## Introduction

Promoting more physical activity (PA) and less sedentary behavior (i.e. sitting or reclining while awake with low energy expenditure; SB) has been a key strategy in the primary and secondary prevention of non-communicable diseases [1]. Although significant efforts have been made in the promotion of PA, about 30% of people still do not meet the required levels of PA recommended in public health guidelines [2, 3]. Moreover, the increasing levels of sedentary behavior across the world and its impact on health and wellbeing is worrisome, given that a large number of adults already spend more than 8 hours per day in SB [4–6].

People with mental disorders are less likely to be sufficiently physically active, compared with the general population [7]. PA promotion has been regarded as a cost-effective strategy in the prevention and management of mental disorders [8]. However, health care professionals, including mental health professionals, find it difficult to implement this strategy in routine clinical consultations and treatment sessions, even when they perceive their clients would benefit from engaging in more PA [9, 10]. Previous studies have found that most health professionals (~60%) do not routinely provide PA counselling to their patients or clients [11]. In an Australian national survey, only 18% of participants reported receiving PA recommendation from their clinician in the past 12 months [12]. It seems this pattern also occurs with mental health professionals as many do not provide PA recommendation to their clients [13, 14].

Several studies have explored the barriers and facilitators experienced by health professionals in recommending PA to their clients [15–17]. The key barriers were a lack of confidence; lack of time; lack of training on how to recommend PA; competing clinical priorities; and perception of health professionals that their clients would not adhere to PA recommendations. The key facilitators were higher levels of PA engagement in health professionals themselves and greater relevant professional knowledge and skills [15–17].

In addition to the identified barriers for health professionals generally, mental health professionals may experience specific barriers to recommending PA, due to their client's mental health conditions, co-morbidities, and adverse effects of medications [16]. This study, therefore, explored the attitudes and practices of mental health professionals in recommending more PA and less SB to their clients, to inform professional education/training and future interventions that can be implemented in mental health services and with the potential to enhance the outcomes of routine mental health treatment.

## Methods

Ethical approval for the "Move More for Mental Health and Wellbeing" study was provided by Victoria University Human Ethics Research Committee [HRE18-123] on 7 August 2018.

Written consent was obtained from the participants during the enrolment of participants into the study.

## Study design and participants

A mixed- method study was chosen as the design for this research which posits a systematic integration of qualitative and quantitative data for a comprehensive and in-depth understanding of attitudes and practices of mental health professionals in recommending more PA and less SB to their clients [18–21]. The quantitative survey enabled us to collect data on a wide range of variables that describe knowledge, attitudes, beliefs, and behaviours of mental health professionals in relation to PA and SB counselling, in the same, standardised way for each participant. In focus group discussions (FGDs), participants could openly discuss perceptions and practices of PA counselling and ways to improve the PA counselling, which enabled us to collect more in-depth information. Results from the quantitative survey and focus groups were triangulated, to explore the research question more deeply and enhance the understanding and trustworthiness of findings [22].

The study was conducted in a convenience sample of 17 mental health professionals who took part in a four-week pre-post intervention trial entitled "Move More for Mental Health and Wellbeing" (clinical trial registration reference: ISRCTN43608761). The study occurred from September to October 2019 in two headspace centres, located in Melbourne, that are part of Australia's national youth mental health service network [23]. These centres provide mental health services to youth aged 12–25 years [24]. We conducted the study by following the Consolidated Standards of Reporting Trials (CONSORT) [25]. Orygen, which is the lead agency of the headspace centres included in the study, provided general oversight of adverse events and participant safety during the study.

## Procedure and measures

Data on attitudes and practices of the participating mental health professionals in providing PA and SB counselling to their clients were collected prior to the intervention using a modified Exercise in Mental Illness Questionnaire (version for health professionals) [26]. In the section on general beliefs, participants were asked to express the level of their agreement with six statements (e.g. "People with a mental illness know that PA is good for their mental health") using a Likert-type response scale with five levels. In the same section, participants were also asked to rank the importance of "increasing PA" and "reducing SB" among a total of 11 treatment strategies for people with mental illness. In the section on perceived barriers for recommending PA to their clients, participants were asked to express their agreement with 11 statements (e.g. "My workload is already too excessive to include recommending PA to people with a mental illness") on a Likert-type response scale with five levels. Participants were then asked how often (on the scale: "Never"; "Occasionally"; "Most of the time"; and "Always") they recommend PA to their clients and how often they recommend reducing SB to their clients. The final set of questions was about specific strategies the participants use to promote PA among their clients. Stanton et al. [26] found that the questionnaire has good measurement properties. Five clinicians and researchers achieved a consensus about the content validity of the questionnaire items [26]. The authors of the current study (including a mental health clinician, an exercise psychologist, a PA epidemiologist, a medical doctor, and an expert in survey design) reached a consensus about the content validity of the modified questionnaire items. The modified items were further reviewed and approved by the lead author of the original questionnaire. For the purpose of this study, we added two items on SB to the questionnaire and used the term "physical activity" instead of "exercise", to capture all types of PA that mental health

professionals can promote among people with mental illness. The test-retest reliability (expressed as intraclass correlation coefficients), of the questions used in this study ranged from 0.61 to 1.00 [26]. The questionnaire also included items about age, gender, marital status, the length of employment in the mental health profession, whether they were a private practitioner/provider or employed by headspace (i.e., salaried staff), whether they had a clinical role at any other service in addition to headspace; whether their headspace role was their main job; and any formal training for recommending PA they may have taken.

This research was based on the social constructivism framework [27, 28]. This framework posits people's beliefs and attitudes are constructed based on their interactions with the social environment [27, 28]. To get a more detailed insight into the subjects of interest, after the intervention, participants were invited to take part in one of two FGDs. Using a semi-structured interview guide, participants for FGDs were asked about: 1) their practices in recommending more PA and less SB to their clients; 2) facilitators for recommending more PA and less SB to their clients; 3) barriers for recommending PA and less SB to their clients; and 4) factors that could improve PA and SB counselling practice (S1 File). The questions in the interview guide were broad and general such that the participants could draw on their experience and practices. Each FGD was attended by five participants and lasted for 45–60 minutes. Each FGD was limited to five participants to ensure that the discussion is sufficiently interactive, inclusive, and participatory and that it allows participants to share in-depth insights in their attitudes and practices. The number of participants per FGD in our study was in accordance with the commonly accepted sample size recommendations for FGDs [29]. The FGDs were facilitated by a lead moderator, while an assistant moderator took field notes and audio-recorded the discussion. At times participants responded to the same question from different perspectives, which resulted in a meaningful discussion within the focus groups. In the second FGD, the themes that evolved overlapped sufficiently enough with the first FGD to assume an adequate saturation of data [30].

### Data analysis

Participants' sociodemographic characteristics and responses to questionnaire items regarding attitudes and practices in recommending PA to people with mental illness were reported using percentages (for categorical data) and means and standard deviations (for numerical data). The analysis was done in version 23 of the IBM Statistical Package for the Social Sciences (SPSS). For the focus group data, interview transcripts and notes were coded and analysed using thematic analysis [31]. The themes were generated through an iterative process of screening and categorising raw data. During the process, the themes were discussed in four meetings, to reach consensus between the researchers (NS, AP and ZP). The participant's responses were categorized into themes based on the principles of realist epistemology [32]. The coding was facilitated using NVivo software, version 12.

## Results

### Sample characteristics

The mean age of participants was 38 years. On average, they had worked in the mental health profession for 8 years. Most of the mental health professionals in the study sample were females, married or in a de facto relationship, and their role in the headspace centre was their main job (76.5% for all). Most of the participants were psychologist (82.3%) and nearly half of participants (47%) had a clinical role at another centre. Most of the participants obtained their highest degree in Australia and were directly employed by the headspace centre (82.3% for

Table 1. Characteristics of respondents.

| Characteristics | Mean ± SD |
|---|---|
| Age | 37.9 ± 9.8 |
| Years in profession | 7.8 ± 7 |
| | Percentage of respondents |
| Male gender | 23.5% |
| Not married | 23.5% |
| Completed their highest degree overseas | 17.7% |
| Non-salaried staff | 17.7% |
| Does not work in a clinical role at any another service | 53.0% |
| Does not consider the role in headspace as the main job | 23.5% |
| Did not undergo formal training in recommending physical activity | 64.7% |
| Role at headspace center | |
| Nurse | 11.8% |
| Psychologist | 70.6% |
| Social worker | 5.9% |
| Clinical lead/psychologist | 11.8% |

both). Around one third of participants (35.3%) had undergone formal training in recommending PA in the treatment of mental illness (Table 1).

## Quantitative findings

Most participants rated the value of recommending PA in the treatment of mental illness as equal or higher than the value of other established treatments (Table 2). Most participants (64.7%) ranked PA counselling among the top three treatment modalities. Only 17.7% of participants ranked SB counselling among the top three treatment modalities (S1 Table). Nearly all participants agreed or strongly agreed that PA is valuable for patients hospitalised with a mental illness in the same manner as outpatients (94.1%). Most participants agreed that mental health benefits of PA for people with mental illness are long lasting (76.4%). Approximately half of participants agreed or strongly agreed that people with a mental illness know that PA is good for their physical health (52.9%) and that people with a mental illness do not engage in PA because they don't think they can (47.1%). Forty-one percent of participants agreed with the statement that people with a mental illness know that PA is good for their mental health.

Table 2. Value of physical activity counselling compared with other types of treatment for mental illness, as perceived by mental health professionals.

| | Significantly less than physical activity | Somewhat less than physical activity | Of equal value to physical activity | Somewhat better than physical activity | Significantly better than physical activity |
|---|---|---|---|---|---|
| Medication | 5.9% | 23.5% | 52.9% | 17.7% | 0% |
| Social support | 0% | 11.8% | 41.2% | 47% | 0% |
| Family therapy | 11.8% | 17.7% | 29.4% | 41.2% | 0% |
| Social skill training | 5.9% | 23.5% | 47% | 23.5% | 0% |
| Cognitive behavioural therapy | 0% | 11.8% | 47% | 41.2% | 0% |
| Vocational rehabilitation | 0% | 29.4% | 58.8% | 11.8% | 0% |
| Electroconvulsive therapy | 17.7% | 35.3% | 23.5% | 23.5% | 0% |
| Bright light therapy | 29.4% | 52.9% | 17.7% | 0% | 0% |

Participants were generally unsure whether people with a mental illness who are recommended PA will adhere to the recommendation or not.

Most participants disagreed or strongly disagreed with all but one statement on barriers for recommending PA as a treatment for mental illness. The most prevalent perceived barriers were a lack of knowledge in PA prescription ("I do not know how to recommend PA to people with a mental illness" and "Prescription of PA to people with mental illness is best delivered by an exercise professional such as an exercise physiologist") and a belief that people with a mental illness will not adhere to a PA program (Table 3).

All participants reported that they recommend PA to their clients at least "occasionally". Eighty-eight percent of them reported at least "occasionally" suggesting to their clients to reduce SB. Personal discussion was the most frequently used strategy to promote PA (88.2%), followed by referral to community-based programs (35.3%) and referral to an exercise physiologist / physiotherapist for recommendation was reported by 17.6% of participants (S2 Table). The most commonly recommended frequency of PA was "As often as you can" and "On most days of the week" (35.3% for both). The most frequently recommended PA intensity for people with mental illness was "At a level that makes you feel good" (47%). One in three participants reported they do not recommend a specific intensity of PA (29.4%). Aerobic exercise was the most commonly prescribed mode of PA (82.4%), followed by team sports (41.2%), and relaxation exercises such as yoga or Tai Chi (41.2%) (S2 Table).

**Table 3. Attitudes and practices of mental health professionals in recommending more PA and less SB to their clients.**

| Beliefs | Strongly disagree | Disagree | Neither disagree nor agree | Agree | Strongly agree |
|---|---|---|---|---|---|
| People with a mental illness know that PA is good for their physical health | 0% | 17.7% | 29.4% | 41.2% | 11.7% |
| People with a mental illness know that PA is good for their mental health | 0% | 29.4% | 29.4% | 41.2% | 0% |
| People with a mental illness do not engage in PA because they don't think they can | 0% | 25.5% | 29.4% | 41.2% | 5.9% |
| PA is valuable for patients hospitalised with a mental illness in the same manner as outpatients | 0% | 0% | 5.9% | 70.6% | 23.5% |
| The physical and mental health benefits of PA for people with a mental illness are not long lasting | 11.7% | 64.7% | 11.7% | 11.7% | 0% |
| People with a mental illness who are recommended PA will not adhere to it | 0% | 29.4% | 41.2% | 29.4% | 0% |
| **Perceived barriers** | | | | | |
| Their mental health makes it impossible for them to participate in PA | 35.3% | 47.0% | 11.7% | 5.9% | 0% |
| I'm concerned PA might make their condition worse | 58.8% | 35.3% | 5.9% | 0% | 0% |
| I am not interested in recommending PA for people with a mental illness | 58.8% | 41.2% | 0% | 0% | 0% |
| I don't believe PA will help people with a mental illness | 64.7% | 35.3% | 0% | 0% | 0% |
| Their physical health makes it impossible for them to participate in PA | 35.3% | 52.9% | 5.9% | 5.9% | 0% |
| I'm concerned they might get injured while engaging in PA | 47.0% | 35.3% | 17.7% | 0% | 0% |
| People with a mental illness won't adhere to a PA program | 29.4% | 23.5% | 23.5% | 23.5% | 0% |
| My workload is already too excessive to include recommending PA to people with a mental illness | 29.4% | 35.3% | 17.7% | 17.7% | 0% |
| Recommending PA to people with a mental illness is not part of my job | 53.0% | 47.0% | 0% | 0% | 0% |
| I do not know how to recommend PA to people with a mental illness | 35.3% | 17.7% | 11.7% | 29.4% | 0% |
| Prescription of PA to people with mental illness is best delivered by an exercise professional such as an exercise physiologist | 23.5% | 11.8% | 35.3% | 23.5% | 5.9% |
| **Practices** | Never | Occasionally | Most of the time | Always | |
| Do you recommend physical activity to people with a mental illness? | 0% | 17.6% | 41.2% | 41.2% | |
| Do you recommend reducing sedentary behaviour (time spent sitting/screen time) to people with a mental illness? | 11.8% | 23.5% | 41.2% | 23.5% | |

## Qualitative findings

In the transcripts from the FGDs, common themes were identified, including: 1) type of recommendations provided to clients; 2) information resources for PA and SB counselling; 3) facilitators for PA and SB counselling; 4) barriers for PA and SB counselling; and 5) factors that could improve PA and SB counselling practice.

**Type of physical activity and sedentary behaviour recommendations provided to clients.**    Participants believed that, as clinicians, they were responsible not only for the mental health but also for the physical health of their clients, and they were interested in incorporating PA and SB counselling routinely in the treatment of mental illness. Although they motivated their clients to move more and spend less time in SB, they tended not to make specific recommendations for type, duration, and frequency of PA or detailed guidance on how to reduce SB. They emphasised that instead of structured exercise, they are more likely to recommend incorporating PA into everyday activities, such as active transport (e.g., walking to the shop instead of driving).

> *"We often talk to young people who spend lot of time sleeping or on their phone and discuss with them what would be the benefit of moving more in general or doing something else that they enjoy, which by default they're going to do more of anyway. So actually, exploring other ways that they can do things which may not necessarily be for the purpose of increasing their movement, but by default, getting them to move more"*
>
> *(Clinician 1).*

Mental health professionals try to create a narrative by explaining to their clients the ways in which increasing PA and reducing SB led to improvements and by providing examples of success stories from other young clients.

> *"I try to create a narrative around 'Why would it be helpful for this specific person at this specific time?'. . . Like, 'It will help in these ways' and 'This is what people said helped from past, but this will also help you."*
>
> *(Clinician 2)*

Mental health professionals also recommended their clients use smartphone applications to increase their motivation for and engagement in PA.

> *"I use different apps; for example, the ones for people interested in getting into running. . . Let's download a running app and make it fun! Or, other things they can use on their phone like pedometers, to make it interactive with their phone, because I guess they are going to be on the phone anyway."*
>
> *(Clinician 3)*

**Information resources for physical activity and sedentary behaviour counselling.**    Participants generally did not have the opportunity to participate in formal education, training, or continued professional development for providing recommendations on PA and SB to their clients. Instead, they relied on other sources of information, such as internet websites and smartphone applications. They also shared knowledge with colleagues and engaged in clinical review meetings and supervision to utilise each other's experience for integrating PA in treating clients.

*"I think just from my own browsing, hearing stuff and also mental health apps."*

*(Clinician 3)*

*"It's just kind of word of mouth."*

*(Clinician 3)*

**Facilitators of physical activity and sedentary behaviour counselling.** Mental health professionals stated that their increased awareness of young people engaging in excessive SB was an important facilitator for recommending more PA and less SB to their clients. There was a shared belief that technology has contributed to increasing the amount of time young people spend in sedentary behaviours, including computer gaming and mobile phone use. Clinicians also perceived that some of their young clients were not as active as they would like to be. This increased awareness was described by clinicians as a motivating factor to discuss possible strategies to increase PA and reduce SB with their clients.

*"There is bit more of awareness around sedentary behaviour and screen time. And I guess that has a kind of flow-on effect of awareness that our clients may be not moving as much as other young people and young people from previous generations. There is a lot more insight on phone and computer activity. I think I have definitely noticed that. Young people seem to have insight into that. But I don't know how much insight they have of the flow-on effects on mental health."*

*(Clinician 3)*

Participants also believed that, if they are more active themselves, they would be in a better position to recommend PA to their clients and share their experiences in maintaining sufficient levels of PA.

*"If you believe in something because you've done it and you also know the evidence base, then it's easy enough to 'sell it'."*

*(Clinician 2)*

*"If I am moving more myself, I can bring it to the work that I do with young people."*

*(Clinician 4)*

**Barriers for physical activity and sedentary behaviour counselling.** Several barriers for recommending more PA and less SB to people with mental illness were identified during the FGDs. For clients with complex mental health needs, clinicians rarely included PA and SB interventions within treatment. In such cases, they generally gave precedence to other types of mental health treatment, such as cognitive behaviour therapy. Moreover, clinicians reported a lack of confidence and competence for providing recommendations on PA and SB to clients with complex clinical needs. Some mental health professionals believed it was inappropriate for them to attend to what may be perceived as less critical or urgent concerns, such as levels of PA, when focusing on the treatment of a mental disorder.

*"I think it just drops off the priority list, even though we may think this would be really helpful."*

*(Clinician 3)*

*"My confidence [for recommending PA] kind of wanes a little bit when there is more pointy mental health stuff going on."*

*(Clinician 1)*

Two barriers stemming from clients' knowledge and perceptions were identified. Mental health professionals stated that young people they work with are typically unaware of the importance of PA for mental health. They also believed that their clients assume that mental health professionals do not have the knowledge and capacity to recommend PA as part of mental health treatment.

*"They visit us with a preconceived idea of coming to sit in room and talk about how they feel and how they can feel better, not necessarily thinking that physical activity is something they could do."*

*(Clinician 1)*

*"This is often seen as just an afterthought rather than what you are going to see a mental health specialist for."*

*(Clinician 5)*

Furthermore, there was a shared perception that including PA in treatment may feel disruptive to clients and potentially damage rapport or the therapeutic alliance. Although participants generally considered PA as a beneficial intervention for mental disorders, their views favoured the need for counselling practices to focus on the presenting issues and concerns of clients.

*"People come to me having been alienated by mental health clinicians trying to make them quit smoking. Like physical activity, quitting smoking can be associated with positive mental health outcomes. But it's not what they came for."*

*(Clinician 6)*

**Factors that could improve physical activity and sedentary behaviour counselling practice.** Participants believed that continued professional development would help them better integrate PA and SB counselling in mental health treatment, with intervention manuals or booklets to present the evidence and provide guidance on how to integrate PA and SB counselling into their therapeutic frameworks specifically identified as helpful to facilitate this.

*"I was going to say professional development. You know we are really interested in getting some manuals for integrating [PA and SB counselling] with cognitive behaviour therapy."*

*(Clinician 7)*

*"...if you can make an obvious link between what they are currently experiencing and how it [PA or SB intervention] will help. So, having that sort of evidence base and rationale would be useful for different sorts of presenting issues."*

*(Clinician 2)*

Participants also thought that education on PA and SB counselling should be a part of formal mental health clinical training. This was discussed as a way to increase the capacity of the

future mental health workforce and broaden the reach and application of PA and SB interventions within mental health services. It was acknowledged, however, that addressing this would require significant investment.

*"I see it as a systemic issue, because people coming out of university have not been told about that important thing, as a component of therapy. Consequently, they are not offering this important component even if they themselves are quite convinced that it should be."*

*(Clinician 5)*

Some mental health professionals also believed that having access to an exercise specialist, such as an accredited exercise physiologist, would benefit their clinical work with young people. An exercise professional could either directly provide exercise interventions to clients or provide professional guidance to mental health clinicians on recommending and integrating PA interventions.

*"It would be amazing to have access to an exercise physiologist, even for a secondary consultation."*

*(Clinician 7)*

## Discussion

The findings of this mixed-methods study indicated that most mental health professionals recognised the benefits of physical activity within mental health treatment, despite a perceived lack of knowledge about and confidence in providing PA and SB counselling to their clients. Mental health professionals considered PA counselling as an important treatment strategy in the treatment of mental illness in young people. However, their assessment of the value of SB counselling was not so favourable. The prevalence of mental health professionals who provide PA and SB counselling "always" or "most of the time" was somewhat higher than that reported in previous studies [13, 33]. This may be because the recommendation for PA has relatively recently been incorporated into the Royal Australian and New Zealand College of Psychiatrists' guidelines for the management of mental disorders [34, 35] and is suggested in the NICE guidelines for youth depression [36]. However, it should be noted that both quantitative and qualitative findings further revealed that clinicians did not provide specific recommendations for PA or SB and made only general suggestions. This might be because these guidelines do not provide such specific instructions on recommending PA recommendation within current treatment frameworks [34, 35]. The current treatment frameworks for people with mental illness can be potentially be improved by incorporating specific recommendations for increasing PA and reducing SB for people with mental illness. Although mental health clinicians are well placed to use the skills they have in behaviour change techniques, findings indicated that they perhaps have not considered how to apply these skills to PA interventions, as almost half of the participants agreed to the statement that people with a mental illness do not engage in PA because they 'don't think they can'.

The fear of potentially disrupting the therapeutic relationship was cited as a major barrier for PA and SB counselling, with the perception that clients would not be interested in receiving such advice when accessing a mental health service and would not adhere to recommendations. However, empirical evidence demonstrates the opposite; clients who received PA recommendation from a clinician were more likely to engage in PA, compared with those who

did not receive such a recommendation [37, 38]. Given that the mental health professionals indicated a need for greater accessibility to the evidence-base to support clinical decisions, the dissemination of the evidence on PA recommendations in mental health treatment needs to be improved using different strategies such as educational seminars, team meetings, and prompt and reminder on the clinical guidelines. This may assist in addressing the identified barriers of lack of resources, competing priorities, lack of knowledge and skills regarding how to and where to find the information [39].

The finding that mental health professionals either do not provide any recommendations about the intensity of PA or link this to the experience of positive affect is consistent with previous studies with health care professionals [40, 41]. Engaging in PA at a self-selected intensity that makes one 'feel good' has been found to have significant, positive effects on physical fitness [42] and mental health [43]. With recent evidence suggesting that apart from PA itself, other associated factors such as enjoyment of the activity, personal preference, choice of activities, and opportunities for social interaction may also be important for mental health benefits [44], mental health professionals should be encouraged to recommend a focus on the enjoyment of physical activities rather than the intensity.

Engaging in structured or unstructured activities, either during leisure time or while commuting, has been identified as important in promoting mental wellbeing [44, 45]. The findings of the present study showed that mental health professionals were more confident in providing recommendations to their clients on unstructured PA that are part of daily living. For recommendations on structured PA, such as exercise, they thought that advice or assistance from an exercise specialist would be helpful. In addition, recommending other resources for PA, such as interactive websites and smartphone apps, were identified as possible strategies. Such apps can provide support, motivation, and guidance to people with mental illness for increasing their leisure time PA.

A previous study found that health professionals who are more physically active are more likely to recommend PA to their clients [46]. Consistent with this finding, the mental health professionals in our study believed that being more physically active themselves would increase their confidence in providing PA counselling to their clients. Such a practical exposure to evidence-informed strategies to increase PA and reduce SB may also address their concerns on how to effectively consult their clients about PA and SB. It may, therefore, be that an intervention to increase PA and reduce SB among mental health professionals would have indirect positive effects on their PA and SB counselling practices. The results of our study are in accordance with the recent findings of Rosenbaum et al. [47] who found improvements in perceived barriers, attitudes, knowledge and confidence in promoting physical health in clients following a lifestyle intervention among clinical and non-clinical mental health staff working in mental health services in Australia.

An important barrier for PA and SB counselling identified in this study was perceived lack of knowledge and skills. Only 35.3% of participants in our study reported having formal PA intervention training and they lacked confidence and perceived competence for providing recommendations on PA and SB to their clients. Mental health professionals in this study reported informally seeking information from online web-based sources or their colleagues. It may not always be easy to locate a reliable source of information for this purpose, and such approach may be too time-consuming for mental health professionals who are already burdened with many other responsibilities. It would therefore be useful to have information on PA and SB counselling when 'on the job' but crucial to ensure it is provided earlier during tertiary training. One of the key perceived barriers was also a concern that PA recommendations may detract from the presenting issues and concerns of clients and potentially damage the therapeutic rapport with clients. Adequate strategies need to be implemented in mental health

centres to help clinicians overcome these perceived barriers and facilitate PA and SB counselling.

As indicated by mental health professionals in our study, the therapeutic role of PA in mental health and PA and SB counselling modules should be considered for inclusion within undergraduate and postgraduate training to ensure greater knowledge and confidence in clinical practice. Similarly, for existing service providers, training should be integrated into continuing professional development to improve knowledge, skills, and attitudes of clinicians regarding PA counselling [48].

Mental health professionals in our study believed that having access to an exercise physiologist would improve the effectiveness of their PA counselling, which is consistent with previous studies [33, 49]. PA counselling may be time consuming and hard to fit in the limited number of routine clinical visits available under current funding arrangements. Receiving assistance from exercise professionals would allow more time for mental health professionals to provide other types of treatment. A study by Ewald et al. [50] found that PA counselling by exercise physiologist for sedentary patients referred by general practitioners were moderately effective. Therefore, a further exploration of multi-disciplinary models of mental health care, including support from PA counsellors or exercise specialists, is warranted.

The strength of this study is its mixed-methods design, which allowed us to gain a deep understanding of mental health professionals' attitudes towards and practices in recommending more PA and less SB to their clients. The limitations of the present study should also be acknowledged. Firstly, this study recruited a small sample of clinicians from youth mental health services. Therefore, they may not be representative of all Australian mental health professionals working across a range of clinical settings. Secondly, the FGDs took place after the mental health professionals received an intervention to increase their PA and reduce SB. Although in the focus group sessions we inquired about their common practices and attitudes prior to the intervention, it may be that the intervention affected some of the responses. It is important to note that this limitation only refers to the qualitative component of the study, because the survey was conducted before the intervention. Researcher interpretation bias is a possible limitation of qualitative research [25]. Reflexivity is considered a crucial component to minimise the possible influence of subjectivity and bias. As a mental health clinician, the focus group moderator (AGP) acknowledges her experience of working withing clinical settings and researching PA in a mental health context. Given that this research specifically explored these constructs, the researchers acknowledge the possibility that her bias towards valuing PA as a mental health intervention may have influenced the way in which she engaged with participants and the responses she elicited. This potential bias was mitigated against by: having another researcher present at the focus groups as an independent observer and note taker; adhering to a semi-structured interview schedule; and having data analysis undertaken by a researcher (NS) who did not moderate the focus groups.

## Conclusion

Although most mental health professionals regularly provide PA and SB counselling to their clients, their recommendations are usually not specific or detailed. PA and SB counselling within mental health treatment could be improved by integrating training on PA and SB counselling in formal education for mental health professionals; providing support from an exercise or PA specialist within a multi-disciplinary approach to mental health care; and implementing interventions to increase PA and reduce SB among mental health professionals themselves. Future studies can build on the findings of this study to specifically explore the impact of providing training to mental health professionals on PA and SB counselling, using

multidisciplinary approaches in PA and SB counselling within mental health care, and increasing PA and reducing SB among mental health professionals themselves.

## Supporting information

**S1 Table. Mental health professionals' perceived importance of different types of treatment for people with mental illness.**
(DOCX)

**S2 Table. Mental health professionals' specific practices in recommending physical activity.**
(DOCX)

**S1 File. Focus group questions.**
(DOCX)

## Acknowledgments

This article is a part of a PhD project of the first author, NS, supervised by AP, SJHB and ZP (principal supervisor).

## Author Contributions

**Conceptualization:** Nipun Shrestha, Zeljko Pedisic, Danijel Jurakic, Stuart J. H. Biddle, Alexandra Parker.

**Data curation:** Nipun Shrestha.

**Formal analysis:** Nipun Shrestha, Zeljko Pedisic, Alexandra Parker.

**Investigation:** Nipun Shrestha, Zeljko Pedisic, Alexandra Parker.

**Methodology:** Nipun Shrestha, Zeljko Pedisic, Danijel Jurakic, Alexandra Parker.

**Project administration:** Nipun Shrestha, Alexandra Parker.

**Supervision:** Zeljko Pedisic, Stuart J. H. Biddle, Alexandra Parker.

**Writing – original draft:** Nipun Shrestha, Alexandra Parker.

**Writing – review & editing:** Nipun Shrestha, Zeljko Pedisic, Danijel Jurakic, Stuart J. H. Biddle, Alexandra Parker.

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
