## [Decision Letter · Decision Letter 0]

5 Mar 2021

PONE-D-20-32740

Physical activity and sedentary behaviour counselling: attitudes and practices of mental health professionals

PLOS ONE

Dear Dr. Parker

Thank you for submitting your manuscript to PLOS ONE. After careful consideration, we feel that it has merit but does not fully meet PLOS ONE’s publication criteria as it currently stands. Therefore, we invite you to submit a revised version of the manuscript that addresses the points raised during the review process.

In addition to addressing the specific queries and comments of the two reviewers, I will request that authors pay close attention to the following points.

In the abstract comment that qualitative data was collected and analysed (also include the analysis approach).In the method section provide sampling approach used; briefly provide the philosophical framework used for the qualitative methodology.Also in the method section provide justification for the use of mixed method; briefly describe the method/procedure/approach used in integration of data from both research methods needed to address the study objective.In the discussion briefly describe moderator’s reflexivity during the focus groups as this important to reduce/declare potential bias in qualitative study/focus group.

We look forward to receiving your revised manuscript.

Kind regards,

Ukachukwu Okoroafor Abaraogu, BMR PT, MSc, PhD

Academic Editor

PLOS ONE

Journal Requirements:

2. Please include additional information regarding the validation of the modified Exercise in Mental Illness Questionnaire used in the study and ensure that you have provided sufficient details that others could replicate the analyses.

Please include copies of the survey questions or questionnaires used in the study, as Supporting Information.

Furthermore, when reporting the results of qualitative research, we suggest consulting the COREQ guidelines: http://intqhc.oxfordjournals.org/content/19/6/349. In this case, please consider including more information on the number of interviewers, their training and characteristics; and please provide additional information regarding the development and pre-testing of the interview guides used as a part of the study.

'The funders had no role in study design, data collection and analysis, decision to publish, or preparation of the manuscript.'

5. We note you have included tables to which you do not refer in the text of your manuscript. Please ensure that you refer to Tables 1 and 3 in your text; if accepted, production will need this reference to link the reader to each Table.

6. Please include captions for your Supporting Information files at the end of your manuscript, and update any in-text citations to match accordingly. Please see our Supporting Information guidelines for more information: http://journals.plos.org/plosone/s/supporting-information

7. Your ethics statement should only appear in the Methods section of your manuscript. If your ethics statement is written in any section besides the Methods, please delete it from any other section.

Reviewers' comments:

Reviewer's Responses to Questions

**Comments to the Author**

1. Is the manuscript technically sound, and do the data support the conclusions?

Reviewer #1: Yes

Reviewer #2: Yes

2. Has the statistical analysis been performed appropriately and rigorously? 

Reviewer #1: Yes

Reviewer #2: Yes

3. Have the authors made all data underlying the findings in their manuscript fully available?

Reviewer #1: No

Reviewer #2: Yes

4. Is the manuscript presented in an intelligible fashion and written in standard English?

Reviewer #1: Yes

Reviewer #2: Yes

5. Review Comments to the Author

Reviewer #1: The authors presented a well-written and structured research report on a mixed-methods approach which explored the attitudes and practices of mental health professional in recommending increment in physical activity levels and reduction of sedentary behaviour to young adults with mental health problems.

The manuscript intends to add to the evidence on the role of mental health professionals in the recommendation of more physical activity and less sedentary behaviour. I also want to highlight positively that the authors’ use of a mixed method provides the opportunity of richer data and findings that may subsequently impact the practice of mental health professionals practice. However, the following minor revisions are recommended to improve the quality of the work presented in this manuscript.

Abstract

1. Not evident that quantitative data was taken

2. Analysis approach was not evident

3. Remove ‘percent’ from line 45

Introduction

4. Remove ‘with’ from the sentence in line 68: ‘given that with a large number of adults already 69 spend more than 8 hours per day in SB [4-6].’

Methods

5. Please state clearly the sampling method used

6. What was the philosophical framework used in this study (required for qualitative methodology)

7. What mixed method design did you use (i.e., sequential explanatory design, sequential exploratory design, triangulation design, embedded design)?

8. Could you please justify the mixed method research design approach (rationale for the type of mixed method design) for this study objective?

9. Is the integration of data from both research methods needed to address the study objective? It appears that data were brought together to form a complete picture at the beginning of the discussion section but how was this done? Guess this is where it may be necessary to explain the mixed method design used.

References for mixed methods design:

Creswell, J., & Plano Clark, V. (2007). Designing and conducting mixed methods research. London: Sage.

O'Cathain, A. (2010). Assessing the quality of mixed methods research: Towards a comprehensive framework. In A. Tashakkori & C. Teddlie (Eds.), Handbook of mixed methods in social and behavioral research (2nd edition) (pp. 531-555). Thousand Oaks: Sage.

Data analysis

10. No allusion to the lead moderator’s reflexivity (journaling) during the focus groups, this is recommended in qualitative research as this may impact the focus group discussions and results, consequently impacting the rigour of the study. Can you comment on this please?

Results

11. Change in line 166: ‘particants’ to ‘participants’

12. Add to line 196: ‘to’ suggesting their clients to in the sentence: ‘Eighty-eight percent of them reported at least “occasionally” suggesting ‘to’ their clients to’

13. Remove space at the beginning of line 216 – space before Participants

14. Table 1: Characteristics of respondents: It is unclear what ‘Completed highest overseas’ mean

Limitation

15. You may need to comment on integration of data if this was not considered in this study.

Reviewer #2: Introduction Section: Line 65, around 30% change to about 30% of people still do not meet the required levels of PA recommended in public health guidelines [2, 3].

Line 68 health and wellbeing is concerning, change to health and wellbeing is worrisome, given that a large number of adults already spend more than 8 hours per day in SB

2.2 Procedure and measures

Line 113-114 rewrite and include counseling; participating mental health professionals in recommending more PA and less SB counseling to their clients

3.2 Quantitative findings

Lines 181 please change the word Around to Approximately half of 182 participants.

4. Discussion

Line 384: In this mixed-methods study, we found that change to It was deduced that

Line 398-400 rewrite to read well.

This might be because these guidelines do not provide such specific instructions on recommending 399 PA recommendation within current treatment frameworks [25, 26] and may be an area 400 for improving the integration of PA and SB counselling within mental health treatment.

Line 431-432: We found that mental health professionals were more confident in providing 432 recommendations to their clients on unstructured PA that are part of daily living.

Change to The findings of the present study showed that mental health professionals were more confident in providing recommendations to their clients on unstructured PA that are part of daily living.

Line 447-450: This hypothesis is supported by the results of a recent study which found improvements in perceived barriers, attitudes, knowledge and 448 confidences in promoting physical health in clients following a lifestyle intervention among clinical and non-clinical mental health staff in mental health treatment settings 450 in Australia [38]. Please include the name of the lead author; a recent study by… and reference appropriately..

Line 452-454: Please when making estimate of number of participants avoid using the around rather be precise.

Line 472 – 474: Mental health professionals in our study believed that having access to an exercise 473 physiologist would improve the effectiveness of their PA counselling, which is 474 consistent with previous studies [24, 40]. Please try as much as possible to acknowledge the work of others when comparing their findings to the results of your research..

5. Conclusion

Line 494: Avoid using the phrase We found

Please rewrite your conclusion, to capture the essence of the research undertakings, do not repeat the results rather highlight important outcome of the study. Mention its implications and significance in solving problems that will benefit the immediate environment where the research was conducted. Suggest ways of improvement in future studies.

General Comments

The manuscript was well written, however, wrong use of comma and other minor grammar issues were observed in the manuscripts. The study sample size was very small and as such the findings should not make too much inference, but claims should acknowledge the limitation of the effect size.

6. PLOS authors have the option to publish the peer review history of their article (what does this mean?). If published, this will include your full peer review and any attached files.

Reviewer #1: No

Reviewer #2: **Yes: **Victor Egwuonwu

---

## [Author Response · Author response to Decision Letter 0]

9 May 2021

Response to reviewers’ comments 

We thank the editor and reviewers for their positive assessment of the manuscript and constructive comments. We addressed all suggestions and provided our point-by-point responses below.

Responses to Academic editor

Academic editor’s comment

In the abstract comment that qualitative data was collected and analysed (also include the analysis approach). 

Author’s response: We appreciate for highlighting this oversight in the reporting of our methods. We have now added the following text: 

Line 35-36: The focus groups were audio-recorded, transcribed, and analysed using thematic analysis.

Academic editor’s comment

In the method section provide sampling approach used; briefly provide the philosophical framework used for the qualitative methodology.

Author’s response: Thank you for the suggestions. We have added the following sentences to the manuscript:

Line 111: The study was conducted in a convenience sample of 17 mental health professionals.

Line 152-154: This research was based on the social constructivism framework. This framework posits people’s beliefs and attitudes are constructed based on their interactions with the social environment. 

Line 159- 161: The questions in the interview guide were broad and general, such that the participants could draw on their experience and practices.

Line 163- 165: At times participants responded to the same question from different perspectives, which resulted in meaningful discussions within the focus groups. 

Academic editor’s comment

Also, in the method section provide justification for the use of mixed method; briefly describe the method/procedure/approach used in integration of data from both research methods needed to address the study objective. 

Author’s response: We thank the editor for highlighting this. We have now added the following justification for the need of mixed methods to address the study objective:

Line 100-110: A mixed-method study was chosen as the design for this research entails a systematic integration of qualitative and quantitative data for a comprehensive and in-depth understanding of attitudes and practices of mental health professionals in recommending more PA and less SB to their clients. The quantitative survey enabled us to collect data on a wide range of variables that describe knowledge, attitudes, beliefs and behaviours of mental health professionals in relation to PA and SB counselling, in the same, standardised way for each participant. In focus group discussions (FGDs), participants could openly discuss perceptions and practices of PA counselling and ways to improve the PA counselling, which enabled us to collect more in-depth information. Results from the quantitative survey and focus groups were triangulated, to explore the research question more deeply and enhance the understanding and trustworthiness of findings. 

Academic editor’s comment

In the discussion briefly describe moderator’s reflexivity during the focus groups as this important to reduce/declare potential bias in qualitative study/focus group.

Author’s response: We have added a description of moderator’s reflexivity during the focus groups and measures taken for its mitigation to the discussion section.

Line 498-508: Researcher interpretation bias is a possible limitation of qualitative research. Reflexivity is considered a crucial component to minimise the possible influence of subjectivity and bias. As a mental health clinician, the focus group moderator (AGP) acknowledges her experience of working within clinical settings and researching PA in a mental health context. Given that this research specifically explored these constructs, the researcher acknowledges the possibility that her bias towards valuing PA as a mental health intervention may have influenced the way in which she engaged with participants and the responses she elicited. This potential bias was mitigated against by: having another researcher present at the focus groups as an independent observer and note taker; adhering to a semi-structured interview schedule; and having data analysis undertaken by a researcher (NS) who did not moderate the focus groups.

Academic editor’s comment

Please include additional information regarding the validation of the modified Exercise in Mental Illness Questionnaire used in the study and ensure that you have provided sufficient details that others could replicate the analyses. Author’s response: According to the editor’s suggestion, we added the following information to the manuscript:

Line 138-139 Five clinicians and researchers achieved a consensus about the content validity of the questionnaire items (ref). 

Line 145-147: The test-retest reliability (expressed as intraclass correlation coefficients), of the questions used in this study ranged from 0.61 to 1.00 (ref).

Line 139-143: The authors of the current study (including a mental health clinician, an exercise psychologist, a PA epidemiologist, a medical doctor, and an expert in survey design) reached a consensus about the content validity of the modified questionnaire items. The modified items were further reviewed and approved by the lead author of the original questionnaire.

Academic editor’s comment

We note that you have indicated that data from this study are available upon request. PLOS only allows data to be available upon request if there are legal or ethical restrictions on sharing data publicly. For information on unacceptable data access restrictions, please see http://journals.plos.org/plosone/s/data-availability#loc-unacceptable-data-access-restrictions.

Author response: Thank you for pointing this out. There are ethical restrictions on sharing the data publicly. The Ethical Committee that approved this study only allowed us to store the data at our university repository, as it may contain sensitive information. To be better aligned with the PLOS policy, we changed the data availability statement.

Line 538-540: The collected data are safely stored in the Victoria University data repository. Controlled data access can be obtained upon a reasonable request submitted to the corresponding author or, in case of her absence, to the Victoria University Human Research Ethics Committee (VUHREC).

Academic editor’s comment

We note you have included tables to which you do not refer in the text of your manuscript. Please ensure that you refer to Tables 1 and 3 in your text; if accepted, production will need this reference to link the reader to each Table.

Author response: Thank you for noticing this. We have now added calls to Table 1 and Table 3 at appropriate places in the manuscript.

Responses to reviewer#1 comments

Reviewer 1 comment:

The authors presented a well-written and structured research report on a mixed-methods approach which explored the attitudes and practices of mental health professional in recommending increment in physical activity levels and reduction of sedentary behaviour to young adults with mental health problems. The manuscript intends to add to the evidence on the role of mental health professionals in the recommendation of more physical activity and less sedentary behaviour. I also want to highlight positively that the authors’ use of a mixed method provides the opportunity of richer data and findings that may subsequently impact the practice of mental health professionals’ practice. However, the following minor revisions are recommended to improve the quality of the work presented in this manuscript. 

Author’s response: Thank you for the positive assessment of our manuscript and constructive comments that helped us improve its quality.

Reviewer 1 comment

Abstract 

Not evident that quantitative data was taken. Analysis approach was not evident

Author’s response: We have now added following statements to emphasize how quantitative data were collected and analysed, as well as how qualitative data were handled.

Line 29-32: Quantitative data were collected using a modified version of the Exercise in Mental Illness Questionnaire in a sample of 17 Australian mental health professionals. The collected data were reported as percentages (for categorical data) and means and standard deviations (for numerical data).

Line 35-36: The focus groups were audio-recorded, transcribed, and analysed using thematic analysis.

Reviewer 1 comment

Remove ‘percent’ from line 45

Author’s response: Thank you for noticing this typo. We have now removed the word “percent”.

Reviewer 1 comment

Introduction

Remove ‘with’ from the sentence in line 68: ‘given that with a large number of adults already spend more than 8 hours per day in SB [4-6].

Author’s response: Thank you for noticing this typo. We have now removed the word “with”.

Reviewer 1 comment

Methods

Please state clearly the sampling method used

Author’s response: We used convenience sampling to recruit mental health professionals to the study, and we added this description in the methods section:

Line 111: The study was conducted in a convenience sample of 17 mental health professionals.

Reviewer 1 comment

What was the philosophical framework used in this study (required for qualitative methodology)

Author response: We used social constructivism framework in our study, and we have added the following sentences to clarify this in the methods section: 

Line 152-154: This research was based on the social constructivism framework. This framework posits people’s beliefs and attitudes are constructed based on their interactions with the social environment.

Reviewer 1 comment

What mixed method design did you use (i.e., sequential explanatory design, sequential exploratory design, triangulation design, embedded design)?

Could you please justify the mixed method research design approach (rationale for the type of mixed method design) for this study objective?

Is the integration of data from both research methods needed to address the study objective? It appears that data were brought together to form a complete picture at the beginning of the discussion section but how was this done? Guess this is where it may be necessary to explain the mixed method design used. Author’s response: Thank you for the suggestion. We have now added the following text to the manuscript:

Line 100-110: A mixed-method study was chosen as the design for this research entails a systematic integration of qualitative and quantitative data for a comprehensive and in-depth understanding of attitudes and practices of mental health professionals in recommending more PA and less SB to their clients. The quantitative survey enabled us to collect data on a wide range of variables that describe knowledge, attitudes, beliefs and behaviours of mental health professionals in relation to PA and SB counselling, in the same, standardised way for each participant. In focus group discussions (FGDs), participants could openly discuss perceptions and practices of PA counselling and ways to improve the PA counselling, which enabled us to collect more in-depth information. Results from the quantitative survey and focus groups were triangulated, to explore the research question more deeply and enhance the understanding and trustworthiness of findings (ref). 

Reviewer 1 comment

Data analysis

No allusion to the lead moderator’s reflexivity (journaling) during the focus groups, this is recommended in qualitative research as this may impact the focus group discussions and results, consequently impacting the rigour of the study. Can you comment on this please?

Author’s response: According to the reviewer’s suggestion, we have now added the following text to the manuscript:

Line 498-508: Researcher interpretation bias is a possible limitation of qualitative research. Reflexivity is considered a crucial component to minimise the possible influence of subjectivity and bias. As a mental health clinician, the focus group moderator (AGP) acknowledges her experience of working within clinical settings and researching PA in a mental health context. Given that this research specifically explored these constructs, the researcher acknowledges the possibility that her bias towards valuing PA as a mental health intervention may have influenced the way in which she engaged with participants and the responses she elicited. This potential bias was mitigated against by: having another researcher present at the focus groups as an independent observer and note taker; adhering to a semi-structured interview schedule; and having data analysis undertaken by a researcher (NS) who did not moderate the focus groups.

Reviewer 1 comment

Results

Change in line 166: ‘particants’ to ‘participants’

Author’s response: Done.

Reviewer 1 comment

Add to line 196: ‘to’ suggesting their clients to in the sentence: ‘Eighty-eight percent of them reported at least “occasionally” suggesting ‘to’ their clients to’

Author’s response: Done.

Reviewer 1 comment

Remove space at the beginning of line 216 – space before Participants

Author’s response: Done.

Reviewer 1 comment

Table 1: Characteristics of respondents: It is unclear what ‘Completed highest overseas’ mean

Author’s response: We have replaced ‘Completed highest overseas’ with “Completed their highest degree overseas”.

Reviewer 1 comment

Limitation

You may need to comment on integration of data if this was not considered in this study

Author’s response: Thank you for pointing this out. We have added a description of integration of data in the methods section.

Responses to reviewer #2 comments

Reviewer 2 comment 

Introduction Section: 

Line 65, around 30% change to about 30% of people still do not meet the required levels of PA recommended in public health guidelines.

Author’s response: Done.

Line 68 health and wellbeing is concerning, change to health and wellbeing is worrisome, given that a large number of adults already spend more than 8 hours per day in SB

Author’s response: Done.

Reviewer 2 comment

Procedure and measures

Line 113-114 rewrite and include counseling; participating mental health professionals in recommending more PA and less SB counseling to their clients

Author’s response: We have made the changes as suggested.

Line 123-124: Data on attitudes and practices of the participating mental health professionals in providing PA and SB counselling to their clients…

Reviewer 2 comment

Quantitative findings

Lines 181 please change the word Around to Approximately half of 182 participants.

Author’s response: Done.

Reviewer 2 comment

Discussion

Line 384: In this mixed-methods study, we found that change to It was deduced that

Author’s response: Done.

Line 398-400 rewrite to read well.

This might be because these guidelines do not provide such specific instructions on recommending PA recommendation within current treatment frameworks [25, 26] and may be an area for improving the integration of PA and SB counselling within mental health treatment.

Author’s response: Thank you for the suggestion. We have now rephrased the sentences as

Line 408-412: This might be because these guidelines do not provide such specific instructions on recommending PA within current treatment frameworks. The current treatment frameworks for people with mental illness can be potentially improved by incorporating specific recommendations for increasing PA and reducing SB for people with mental illness.

Reviewer 2 comment

Line 431-432: We found that mental health professionals were more confident in providing recommendations to their clients on unstructured PA that are part of daily living. Change to The findings of the present study showed that mental health professionals were more confident in providing recommendations to their clients on unstructured PA that are part of daily living.

Author’s response: Thank you. We have made the changes as suggested.

Reviewer 2 comment

Line 447-450: This hypothesis is supported by the results of a recent study which found improvements in perceived barriers, attitudes, knowledge and confidences in promoting physical health in clients following a lifestyle intervention among clinical and non-clinical mental health staff in mental health treatment settings in Australia [38]. Please include the name of the lead author; a recent study by… and reference appropriately.

Author’s response: We have made the changes as suggested.

Line 454-458: The results of our study are in accordance with the recent findings of Rosenbaum et al. [45], who found improvements in perceived barriers, attitudes, knowledge, and confidence in promoting physical health in clients following a lifestyle intervention among clinical and non-clinical staff working in mental health services in Australia.

Reviewer 2 comment

Line 452-454: Please when making estimate of number of participants avoid using the around rather be precise.

Author’s response: We have made the changes as suggested.

Line 460: Only 35.3% of participants in our study reported having formal PA intervention training.

Reviewer 2 comment

Line 472 – 474: Mental health professionals in our study believed that having access to an exercise physiologist would improve the effectiveness of their PA counselling, which is consistent with previous studies [24, 40]. Please try as much as possible to acknowledge the work of others when comparing their findings to the results of your research.

Author’s response: We have made the changes as suggested and added following sentences for more clarity.

Line 484 – 485: A study by Ewald et al. [48] found that PA counselling provided by exercise physiologists for sedentary patients referred by general practitioners was moderately effective.

Reviewer 2 comment

Conclusion

Line 494: Avoid using the phrase We found

Author’s response: We have made the changes as suggested.

Reviewer 2 comment

Please rewrite your conclusion, to capture the essence of the research undertakings, do not repeat the results rather highlight important outcome of the study. Mention its implications and significance in solving problems that will benefit the immediate environment where the research was conducted. Suggest ways of improvement in future studies.

Author’s response: We have rewritten the conclusion as suggested. 

 Line 511-520: Although most mental health professionals regularly provide PA and SB counselling to their clients, their recommendations are usually not specific or detailed. PA and SB counselling within mental health treatment could be improved by integrating training on PA and SB counselling in formal education for mental health professionals; providing support from an exercise or PA specialist within a multi-disciplinary approach to mental health care; and implementing interventions to increase PA and reduce SB among mental health professionals themselves. Future studies can build on the findings of this study to specifically explore the impact of providing training to mental health professionals on PA and SB counselling, using multidisciplinary approaches in PA and SB counselling within mental health care, and increasing PA and reducing SB among mental health professionals themselves. 

Reviewer 2 comment

General Comments

The manuscript was well written; however, wrong use of comma and other minor grammar issues were observed in the manuscripts. The study sample size was very small and as such the findings should not make too much inference, but claims should acknowledge the limitation of the effect size.

Author’s response: Thank you for pointing this out. We have thoroughly checked the manuscript and corrected grammatical errors. We have also acknowledged the limitation of the small sample size in the limitations section.

---

## [Decision Letter · Decision Letter 1]

23 Jun 2021

PONE-D-20-32740R1

Physical activity and sedentary behaviour counselling: attitudes and practices of mental health professionals

PLOS ONE

Dear Dr. Parker,

Thank you for submitting your manuscript to PLOS ONE. After careful consideration, we feel that it has merit but does not fully meet PLOS ONE’s publication criteria as it currently stands. Therefore, we invite you to submit a revised version of the manuscript that addresses the points raised during the review process.

Please provide justification for the sample size (or whether data satisfaction has been reached) and acknowledge the limitation of the small sample size of the study.

We look forward to receiving your revised manuscript.

Kind regards,

Rainbow T. H. Ho

Academic Editor

PLOS ONE

Journal Requirements:

Additional Editor Comments (if provided):

Please provide justification for the sample size (or whether data satisfaction has been reached) and acknowledge the limitation of the small sample size of the study.

Reviewers' comments:

Reviewer's Responses to Questions

**Comments to the Author**

1. If the authors have adequately addressed your comments raised in a previous round of review and you feel that this manuscript is now acceptable for publication, you may indicate that here to bypass the “Comments to the Author” section, enter your conflict of interest statement in the “Confidential to Editor” section, and submit your "Accept" recommendation.

Reviewer #1: All comments have been addressed

Reviewer #2: All comments have been addressed

2. Is the manuscript technically sound, and do the data support the conclusions?

Reviewer #1: Yes

Reviewer #2: Yes

3. Has the statistical analysis been performed appropriately and rigorously? 

Reviewer #1: Yes

Reviewer #2: Yes

4. Have the authors made all data underlying the findings in their manuscript fully available?

Reviewer #1: No

Reviewer #2: Yes

5. Is the manuscript presented in an intelligible fashion and written in standard English?

Reviewer #1: Yes

Reviewer #2: Yes

6. Review Comments to the Author

Reviewer #1: Thank you for the detailed responses to the reviewers’ comments. The manuscript is much improved, and ready for acceptance for publication.

Reviewer #2: The manuscript was painstakingly written and we'll presented by the authors. I recommend that the manuscript be published by PLOS ONE.

7. PLOS authors have the option to publish the peer review history of their article (what does this mean?). If published, this will include your full peer review and any attached files.

Reviewer #1: No

Reviewer #2: **Yes: **Afamefuna Victor EGWUONWU

---

## [Author Response · Author response to Decision Letter 1]

29 Jun 2021

Dear Editors,

Thank you for your suggestions. We have made the additional changes to the manuscript PONE-D-20-32740R1 that were recommended. Please find our point-by-point responses to your suggestions below.

Thanking you for your consideration. 

Yours sincerely,

Authors

Response to Editor’s comments

Editor’s comment: Please provide justification for the sample size (or whether data satisfaction has been reached) and acknowledge the limitation of the small sample size of the study.

Authors’ response: 

According to your suggestions, we have added the following sentences to the manuscript:

Lines 167-170: Each FGD was limited to five participants to ensure that the discussion is sufficiently interactive, inclusive, and participatory and that it allows participants to share in-depth insights in their attitudes and practices. The number of participants per FGD in our study was in accordance with the commonly accepted sample size recommendations for FGDs [29].

Lines 173-175: In the second FGD, the themes that evolved overlapped sufficiently enough with the first FGD to assume an adequate saturation of data [30].

Thank you for including your ethics statement on the online submission form: "The ethical approval for the “Move More for Mental Health and Wellbeing” was provided by Victoria University Human Ethics Research Committee [HRE18-123] on 07/08/2018. Written consent was obtained from the participants during the enrolment of participants into the study.".

To help ensure that the wording of your manuscript is suitable for publication, would you please also add this statement at the beginning of the Methods section of your manuscript file.

This statement has been added (Lines 99-102).

---

## [Editor Report · Decision Letter 2]

1 Jul 2021

Physical activity and sedentary behaviour counselling: attitudes and practices of mental health professionals

PONE-D-20-32740R2

Dear Dr. Parker,

We’re pleased to inform you that your manuscript has been judged scientifically suitable for publication and will be formally accepted for publication once it meets all outstanding technical requirements.

Kind regards,

Rainbow T. H. Ho

Academic Editor

PLOS ONE

Additional Editor Comments (optional):

Can consider to remove the repeated information of ethical approve in line 121.
---

## [Editor Report · Acceptance letter]

8 Jul 2021

PONE-D-20-32740R2 

Physical activity and sedentary behaviour counselling: attitudes and practices of mental health professionals 

Dear Dr. Parker:

I'm pleased to inform you that your manuscript has been deemed suitable for publication in PLOS ONE. Congratulations! Your manuscript is now with our production department. 

Kind regards, 

on behalf of

Professor Rainbow T. H. Ho 

Academic Editor

PLOS ONE